# ADDING 32 PARAMETERS TO A LLM CAN IMPROVE FINE-TUNED CLASSIFICATION PERFORMANCE BY UP TO 1.5-6 PERCENTAGE POINTS

## ABSTRACT

In this paper, we introduce and analyze an architectural augmentation for Large Language Models (LLMs) that enhances their performance in fine-tuned classification tasks through a minimalistic yet effective approach. By incorporating one additional learnable parameter per transformer block, we facilitate a depth-wise pooling mechanism that leverages the hierarchical information encoded in the layers of the pre-trained model. We demonstrate that this method significantly improves classification accuracy, with an observed increase of 1.5-6 percentage points in some models, at a negligible compute cost during training. Our experiments span various models and datasets, underscoring the universality and adaptability of the proposed technique. The augmentation proves particularly effective under data-scarce conditions, highlighting its practical utility in real-world applications. The simplicity and efficacy of our approach advocate for its integration in fine-tuning LLMs for classification, promising enhanced performance and new insights into model interpretability and optimization for diverse NLP tasks.

## 1 INTRODUCTION

Ignited by the release of the Transformer paper (Vaswani et al., 2017) and fueled by many important subsequent works (Devlin et al., 2018; Liu et al., 2019b; Radford et al., 2018; 2019), powerful pre-trained Large Language Models (LLMs) have become an indispensable part of the natural language processing (NLP) landscape for researchers and practitioners alike.

LLMs are typically pre-trained on extensive text corpora using methods such as autoregressive modeling or Masked Language Modeling (MLM). They are then fine-tuned for specific tasks, with the final layer often being customized to suit the target application. However, this approach potentially underutilizes the rich, hierarchical information embedded within the multiple layers of the model. Each layer captures unique and valuable insights (Raganato & Tiedemann, 2018; Tenney et al., 2019; Voita et al., 2019), and solely relying on the final layer could limit the model's performance potential.

In this paper, we offer further analysis of the problem, and address it via a minimal architectural augmentation aimed at harnessing the depth-wise information within the LLMs. Our objective is to enhance the model's performance in classification tasks without introducing significant complexity or computational overhead.

In the proposed method one additional learnable parameter per transformer block is added. This facilitates a depth-wise pooling mechanism, allowing the model to leverage the information encoded across various layers. We hypothesize that this approach will enable a more nuanced and comprehensive utilization of the model's pre-trained knowledge, leading to an enhanced performance in fine-tuned classification tasks. Furthermore, analyzing the trained weights of P-32 will provide additional information regarding the efficacy of information at different levels of depth in the hierarchy. We named our method P-32, since for most LLM architectures we only add 32 parameters to the model.

We demonstrate that P-32 can significantly improve classification accuracy, with an observed increase of up to almost 6 percentage points in the best performing model. Our experiments, conducted across various models and datasets, underscore the universality and adaptability of our technique. We

provide an in-depth analysis of the learned depth-wise weights, offering insights into the model's enhanced capacity to utilize its hierarchical knowledge effectively.

Our key contributions include:

- We propose an architectural augmentation that improves classification performance by fully utilizing the hierarchical information captured by the pre-trained Large Language Model (LLM).
- We show that the proposed architecture are able to select and retain task relevant information captured at different depths in the LLM.
- We show how the proposed method improves performance across a plethora of different model sizes and in tandem with popular fine-tuning techniques.
- We show that the proposed method specifically and successfully addresses the problem of overthinking.

The remainder of this paper is structured as follows: First, we explore related work in Section 2 and subsequently introduce the proposed method in Section 3. Section 4 provides empirical results, whilst Section 5 focuses on analyzing the factors responsible for the superior performance. Finally, we conclude in Section 7

## 2 RELATED WORK

In this section, we provide an overview of the relevant literature in relation to our proposed methodology. This includes the development of the Transformer architecture, large pre-trained language models, and various techniques for fine-tuning and augmentation of these models for specific tasks.

### 2.1 TRANSFORMER ARCHITECTURE

The Transformer architecture was introduced by Vaswani et al. in the paper "Attention is All You Need" (Vaswani et al., 2017). This groundbreaking work demonstrated that self-attention mechanisms could be used to efficiently model long-range dependencies in sequence data, outperforming previous state-of-the-art models based on recurrent neural networks (RNNs) and convolutional neural networks (CNNs) in various NLP tasks. Since then, the Transformer architecture has been the foundation for numerous works in the NLP domain.

### 2.2 LARGE PRE-TRAINED LANGUAGE MODELS

The concept of pre-training language models on vast amounts of text data before fine-tuning them for specific tasks has been a driving force behind the recent progress in NLP. Notable models include BERT (Devlin et al., 2018), GPT (Radford et al., 2018; 2019), RoBERTa (Liu et al., 2019b), and many others. These models have demonstrated remarkable performance across a range of tasks, such as question-answering, sentiment analysis and text summarization, often surpassing human performance.

### 2.3 FINE-TUNING AND MODEL AUGMENTATION

Several innovative strategies have been put forth to fine-tune and boost the effectiveness of pre-trained language models. One such strategy involves the use of intermediate layer representations (Houlsby et al., 2019). This approach allows a LLM to be fine-tuned for a variety of tasks without fine-tuning the weights of the Large Language Model.

Another method to enhance performance is the introduction of task-specific adaptation layers (Stickland & Murray, 2019). In this scenario, additional layers are incorporated into the model that are specifically designed and trained to optimise performance for a particular task. This allows the model to maintain its general language understanding capabilities while also excelling in a specific task, creating a balance between generalisation and specialisation.

The strategy of multi-task learning also shows considerable promise (Liu et al., 2019a). This approach trains a single model on multiple related tasks simultaneously, with the idea that learning from one

task can inform and improve the model's performance on others. This is based on the principle of 'transfer learning', where knowledge gained while solving one problem is applied to a different but related problem.

Lastly, the inclusion of external knowledge sources has demonstrated beneficial results (Peters et al., 2019; Zhang et al., 2019). In this approach, additional information from outside the model's training data is incorporated into the learning process. This could be in the form of structured knowledge bases, unstructured text data, or even real-time information. The additional knowledge can provide valuable context, help disambiguate meanings, and add depth to the model's understanding, thereby enhancing its overall performance.

### 2.4 EARLY-EXIT AND MULTI-EXIT BERT ARCHITECTURES

Early-exit or multi-exit BERT architectures have emerged to address computational costs in low-latency and resource-constrained environments, enabling early classification based on intermediate layer representations (Xin et al., 2020; Zhou et al., 2020).

## 3 MODEL

Our proposed augmentation is designed to enhance the fine-tuning performance of Large Language Models (LLMs) by introducing a learned, depth-wise pooling mechanism. This method addresses the "overthinking" issue (Kaya & Dumitras, 2018), where networks, especially deep ones, can misclassify simpler samples due to excessive processing.

Before giving a technical introduction to the method, we offer an intuitive motivation: LLMs learn different levels of representation at various depths (Raganato & Tiedemann, 2018; Tenney et al., 2019; Voita et al., 2019). Lower layers often capture basic features like syntax and grammar, while deeper layers encapsulate more complex, abstract features such as semantics and context. However, not all tasks require the same level of abstraction. Some might benefit more from the concrete, syntactical representations found in the shallower layers, while others need the nuanced, abstract features from the deeper layers. Our method allows the model to select and combine these diverse representations.

In our P-32 method, during the forward pass, the hidden states from each transformer block are collected and pooled using a weighted sum, as shown in Equation 1. The weights are determined through a learnable parameter associated with each block, facilitated by the Softmax function. This adaptability ensures that the most relevant features are selected and combined, tailored to the specific task at hand.

$$\tilde{h} = \sum_{i \in H} h_i * \left( \frac{e^{\lambda_i}}{\sum_j e^{(\lambda_j)}} \right) \tag{1}$$

Where,

- $h_i$ is the hidden state at depth $i$
- $\lambda_i$ is the learned parameter (a scalar) for $h_i$
- $H$ is the set of all hidden states.

Figure 1 shows two diagrams the left one is a standard LLM classification setup, whilst the one on the right illustrates the P-32 method, where hidden states from each transformer block are pooled using the learned weights. This pooled representation, encapsulating features from various depths, is then fed through the standard classification head.

## 4 EXPERIMENTS

To fairly assess the empirical performance of the proposed method, we selected four language models that we deemed representative of most language models in terms of size, pre-training strategy and pre-training dataset. Solely based on its popularity with practitioners, we include the recently released LLAMA-2-{7,13}B models (Touvron et al., 2023); BERT, as a model representing alternative

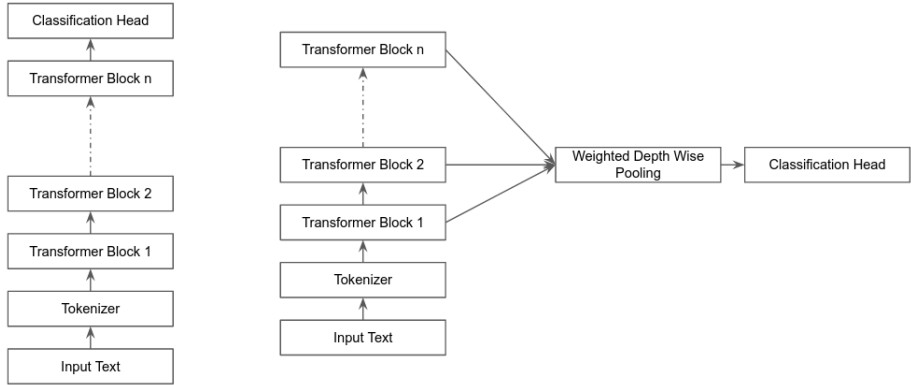

Figure 1: Diagrams for a normal LLM classification setup (left) and P-32 (right).

pretraining objectives (namely Masked Language Modeling (MLM) and Next Sentence Prediction (NSP)) (Devlin et al., 2018), and lastly, we use StableCode-Instruct-Alpha (Adithyan et al.). The reasoning behind the last choice was to include a model that is trained not solely on text.

Table 1 summarizes some basic information regarding the size and number of transformer blocks each of the language models use.

| Model | # Transformer Blocks | # Parameters |
|---|---|---|
| BERT | 12 | 109,482,240 |
| StableCode | 32 | 2,769,310,720 |
| LLAMA-2-7B | 32 | 6,738,415,616 |
| LLAMA-2-13B | 40 | 13,015,864,320 |

Table 1: The number of transformer blocks (# Transformer Blocks) and total parameter count (# Parameters) of the used models.

Since the GLUE benchmark consists of a number of different datasets of different sizes, it seems like a sensible choice for checking how P-32 addresses overthinking on smaller datasets, whilst not degrading performance on larger ones. Having said that, naturally, we do not expect the, for example, LLAMA-2 models to outperform BERT on this benchmark, but rather are curious to compare the baseline versions of each with the relevant P-32 version.

Since the parameter count in modern Large Language Models has vastly out-paced the amount of affordable compute available, we will use QLoRA (Dettmers et al., 2023) for all models, except BERT.

### 4.1 MAIN RESULTS

Before diving into the results, we want to point out that due to the high cost of fine-tuning large language models, we were only able to complete a single run for each of them. However, since the results seem very coherent, we don't think that this is an issue. Additionally, when training the LLAMA models on the smaller datasets in GLUE, the model can sometimes diverge into a nan loss. If that happens, we change the seed and re-run the model (worth pointing out that this only happend once or twice, and then mainly for the baseline versions, and not for the P-32 ones).

As can be seen in Table 2, our method consistently improves the performance of all models, except BERT, where it degrades it by 0.02 percentage points. This is in-line with what we would expect as the relatively shallow BERT architecture is less likely to suffer from overthinking than the deeper and larger models. The largest performance gains are achieved when applying the P-32 method to StableCode, with a 5.85 percentage point increase. This is likely because the model was originally trained on a combination of code and text, whilst the benchmark datasets are text based. Thus it is forced to switch domains to an extend, which seems to be made easier via the depth-wise pooling;

| | BERT | | StableCode | | LLAMA-2-7B | | LLAMA-2-13B | |
|---|---|---|---|---|---|---|---|---|
| | Baseline | P-32 (ours) | Baseline | P-32 (ours) | Baseline | P-32 (ours) | Baseline | P-32 (ours) |
| rte | 62.46 | **65.34** | 55.96 | **58.48** | 54.87 | **57.04** | 49.10 | **53.79** |
| mrpc | **88.62** | 87.77 | 79.80 | **79.66** | 80.47 | **83.91** | **83.31** | 80.89 |
| cola | **56.53** | 55.74 | 5.13 | **13.20** | 46.40 | **50.89** | 47.93 | **55.16** |
| stsb | 88.52 | **88.65** | 56.61 | **70.73** | 84.88 | **87.20** | 83.92 | **86.30** |
| sst2 | **92.55** | 91.28 | 83.83 | **85.09** | **94.90** | 94.38 | 94.61 | **94.95** |
| qnli | 90.87 | **90.98** | 70.60 | **79.99** | **89.01** | 88.82 | 91.21 | **91.29** |
| qqp | **87.52** | 87.37 | 83.04 | **83.21** | **87.06** | 85.07 | **87.43** | 87.40 |
| mnli | 84.12 | **84.19** | 62.42 | **71.83** | 80.85 | **85.75** | **87.35** | 87.28 |
| mnli (mm) | **84.42** | 84.11 | 64.25 | **72.09** | 82.90 | **86.25** | 87.45 | **87.61** |
| Avg. | **81.73** | 81.71 | 62.40 | **68.25** | 77.93 | **79.92** | 79.15 | **80.52** |

Table 2: The GLUE performances of four different language models by themselves and when augmented with P-32. Scores are reported as Accuracy for all, except: mrpc (F1), qqp (F1) and stsb (Spearman-Correlation). The datsets are ordered by increasing size.

likely because it is not required to change the weights throughout the network, but just changing the first few layers will already have a large impact.

Beyond simply comparing the performance of our method to the baseline, we want to further analyze what pooling strategy is learned (Section 5.2), how different dataset sizes influence the model performance (Section 5.3), and for which type of datasets P-32 is worth using (Section 5.4).

# 5 ANALYSIS

## 5.1 ON THE PROBLEM OF OVERTHINKING

In Section 4 we argued that P-32 does not lead to significant performance gains on BERT because the network is likely too small to be suffering from overthinking. To substantiate this claim, we train BERT multiple times on GLUE, each time only using the first n transformer blocks. If indeed the network does not suffer from overthinking, we expect to scores to strictly increase with depth.

| Num Blocks | 3 | 6 | 9 | 12 |
|---|---|---|---|---|
| rte | 57.40 | **64.26** | 62.46 | 62.46 |
| mrpc | 80.32 | 82.38 | 86.29 | **88.62** |
| cola | 5.71 | 31.00 | 51.53 | **56.53** |
| stsb | 59.57 | 85.74 | 87.42 | **88.52** |
| sst2 | 88.53 | 88.76 | 91.63 | **92.55** |
| qnli | 81.93 | 86.80 | 88.19 | **90.87** |
| qqp | 82.62 | 86.27 | 86.61 | **87.52** |
| mnli | 74.13 | 80.43 | 82.01 | **84.12** |
| mnli (mm) | 74.65 | 80.36 | 82.31 | **84.42** |
| Avg. | 67.21 | 76.33 | 79.83 | **81.73** |

Table 3: The performance achieved by BERT when only using the first n transformer blocks. Scores are reported as Accuracy for all, except mrpc (F1), qqp (F1) and stsb (Spearman-Correlation).

As can be seen in Table 3 the scores are indeed strictly increasing with network depth for all datasets, except the smallest one: rte. This means that BERT only sufferes from overthinking for rte, but not for the other datasets. Coincidentally, as can be seen in Table 2, rte is the only dataset for which using P-32 with BERT leads to a significant performance gain (2.88 percentage points). Though it is impossible to conclude for sure, we see this as extremely strong evidence that P-32 does indeed address the problem of overthinking.

Next, it is worth checking if LLAMA-2-7B, is overthinking any of the glue datasets. To that end, we follow the same set-up as above, only training the first n transformer blocks, and show the achieved performances in Table 4.

| Num Blocks | 5 | 10 | 15 | 20 | 25 | 30 |
|---|---|---|---|---|---|---|
| rte | 52.71 | 56.32 | 54.51 | **62.82** | 48.74 | 52.34 |
| mrpc | 77.52 | 82.02 | 84.45 | 81.07 | 80.36 | **84.5** |
| cola | 20.99 | 46.11 | 46.73 | **49.23** | 36.69 | 47.25 |
| stsb | 78.00 | 78.66 | 85.21 | **85.71** | 85.12 | 84.27 |
| sst2 | 86.35 | 93.23 | 94.15 | **95.07** | 94.61 | 94.72 |
| qnli | 75.93 | 86.89 | **89.69** | 89.6 | 88.34 | 88.94 |
| qqp | 81.54 | 85.43 | 86.66 | 86.57 | 86.73 | **86.88** |
| mnli | 71.85 | 81.4 | 85.84 | 85.17 | **85.9** | 84.42 |
| mnli (mm) | 71.75 | 82.06 | 86.42 | 86.49 | **86.54** | 85.1 |
| **Avg.** | **68.52** | **76.90** | **79.30** | **80.19** | **77.00** | **78.71** |

Table 4: The performance achieved by an LLAMA-2-7B model when only using the first n transformer blocks. Scores are reported as Accuracy for all, except mrpc (F1), qqp (F1) and stsb (Spearman-Correlation). The corresponding visualization can be found in Appendix A

In most datasets, there is a notable trend where performance initially rises with an increase in the number of transformer blocks, only to decline thereafter. Thus we can see that overthinking does indeed present a problem to LLAMA-2-7B across most GLUE datasets. Additionally, the optimal number of transformer blocks used seems to differ by dataset. Therefore, there is no one-fits-all solution; indicating that allowing the network to learn how much information to use from each layer can indeed be beneficial.

## 5.2 LEARNED DEPTH-WISE WEIGHTS

The aspect of our proposed model that is most interesting to analyze, are the learned weights for depth-wise pooling. To that end, we train a LLAMA-2-7B model on each dataset of the GLUE benchmark and extract the learned weights.

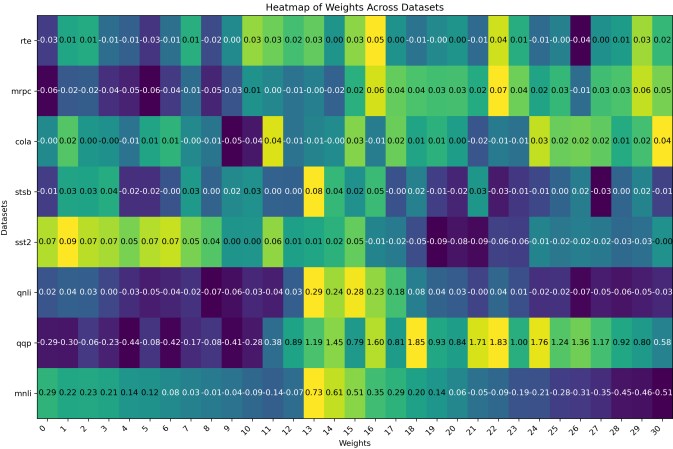

Figure 2: The raw (not passed through softmax) learned depth-wise pooling weights for each dataset visualized as a row-wise normalized heatmap.

Once again, we have sorted the datasets in ascending order from top to bottom by the number training samples. Figure 2 offers a few expected and unexpected insights. Firstly, there is a general trend where the larger a dataset is, the less importance is given to the first few transformer blocks (for example, see qqp, qnli, mnli), whilst there are a number of smaller datasets where a lot of attention is paid to the first few transformer blocks, most notably, sst2. Additional information regarding the size of the various datasets can be found in Table 5.

Going beyond the general trends, using the information from Figure 2 in combination with the results in Table 4 (where we only used the first n transformer blocks of LLAMA-2-7B) should tell us whether the model did indeed learn to pull information from layers that are most beneficial towards the correct classification. Most pronounced amongst these comparisons is qnli. In Table 4 we identified 15 transformer blocks to work best, and indeed, in P-32, the highest weights are associated with transformer blocks 14, 15, 16 and 17; whilst weights towards the lower and higher layers are much smaller. However, such a clear connection cannot always be drawn. For sst2, for example, the weights learned (highest for the first few layers), do not correspond to the number of transformer blocks that worked best in our previous experiment (20 transformer blocks). It is worth pointing out that of course, in Table 4 we are comparing test performances, whilst the model, when learning it's weights is simply trying to minimize the training loss.

Lastly, it is interesting to see that when the model is trained on the larger datasets, the P-32 weights have a more or less smooth distribution (especially when compared to the smaller dataset). This might suggest that the weights have not converged for the smaller ones, a problem no doubt caused by the low learning rate and aggressive gradient clipping used when fine-tuning a LLM. A future avenue of work that is worth exploring is assigning a different learning rate to the P-32 weights.

## 5.3 THE IMPACT OF THE AMOUNT OF DATA

As mentioned above, in this section we will explore the impact different sized datasets have on the performance of our method. To that end, we train the LLAMA-2-7B model (both with and without P-32) on 20, 40, 60, 80 and 100% of the training data from the stsb dataset.

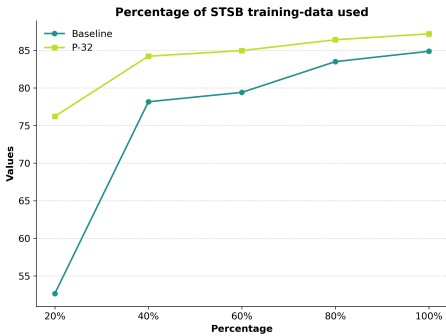

Figure 3: The performance of a fine-tuned LLAMA-2-7B model both without and with P-32, when only using a fraction of the available training data of the stsb dataset. Results reported are Spearman Correlation Scores.

As can be seen in Figure 3, when adding the proposed depth-wise pooling, the model does not only perform better, but as we use less and less data, the gap between the baseline and P-32 widens. This is a good indication that the proposed method works better under data scarcity than the baseline. We also want to mention that since the large drop-off in performance for the baseline model at 20% of training data seemed un-natural, we confirmed that it is the actual result by re-running that part of the experiment three times with three different seeds. However, all three experiments returned similar values.

This result in combination with the learned weight patterns for mnli & qqp (See Figure 2 makes us hypothesize that our method performs well on smaller datasets by utilizing the first few layers of the network, but when trained for longer on a larger dataset, learns to utilize later layers in the network.

## 5.4 ANALYSIS OF DATASETS

In this section, we delve deeper into the statistical relationships between the characteristics of datasets and the performance gain observed from using the augmented language model. The experiments are

run on the LLAMA-2-7B model. We conducted an exhaustive analysis, focusing on two key dataset features: the number of training samples and the average input length (See Table 5).

| Dataset | # Train Samples | Avg. Input Length | Type |
|---------|-----------------|-------------------|------|
| cola | 8,551 | 7.70 | Binary |
| sst2 | 67,349 | 9.41 | Binary |
| mrpc | 3,668 | 43.89 | Binary |
| stsb | 5,749 | 9.94 | Continuous |
| qqp | 363,846 | 11.06 | Binary |
| mnli | 392,702 | 14.89 | Ternary |
| qnli | 104,743 | 26.52 | Binary |
| rte | 2,490 | 26.18 | Binary |

Table 5: For each of the GLUE datasets we list the number of training samples (# Train Samples), the average input length (Avg. Input Length), and the class type (Type).

### 5.4.1 TRAINING SAMPLES VS PERFORMANCE GAIN

The relationship between the number of training samples and the performance gain was examined using a linear regression model. The model yielded a coefficient of -1.428e-05, indicating a negative correlation between the two variables. In other words, as the number of training samples increases, the performance gain of the augmented model tends to decrease (the coefficient is so small because the # Train Samples are in the thousands, whilst the performance gain is at most a single digit value).

The R-squared value of 0.644 ($p = 0.030$) suggests that approximately 64.4% of the variability in performance gain can be explained by the number of training samples. This substantial proportion underscores the significance of the dataset size in influencing the efficacy of the augmented model.

### 5.4.2 AVERAGE INPUT LENGTH VS PERFORMANCE GAIN

We also investigated the impact of the average input length on performance gain. The linear regression model produced a coefficient of 0.0425, signifying a positive, though weak, correlation. This suggests that datasets with longer average input lengths are associated with slightly higher performance gains when using the augmented model.

However, the R-squared value is 0.060, with a p-value of 0.596, thus we cannot conclude that the average input length has a statistically significant impact on the success of applying P-32.

### 5.4.3 INTERPRETRATION AND IMPLICATIONS

Our interpretation of the negative correlation between the number of training samples and performance gain is that the larger the dataset is, the less likely the network is to overthink, thus the performance gain achieved by P-32 shrinks relative to the baseline. Because of the potentially large performance gains, fast implementation and low computational cost, we advice practitioners to use P-32 when fine-tuning LLMs for classification with datasets that are smaller than 100,000 elements.

## 6 ABLATION

In this ablation study we compare a LLAMA-2-7B augmented with the P-32 method (denoted as P-32) to one that uses simple depth-wise average pooling (denoted as Avg.Pool; i.e. Equation 2).

$$\tilde{h} = \frac{1}{I} \sum_{i \in H} h_i \tag{2}$$

Where,

- $h_i$ is the hidden state at depth $i$
- $I$ is the total number of transformer blocks

- $H$ is the set of all hidden states.

For reference, we will also include the baseline results (denoted as Baseline).

| LLAMA-2-7B | Baseline | P-32 | Avg.Pool |
|---|---|---|---|
| rte | 54.87 | **57.04** | 51.63 |
| mrpc | 80.47 | **83.91** | 76.79 |
| cola | 46.40 | 50.89 | **51.76** |
| stsb | 84.88 | **87.20** | 83.85 |
| sst2 | 94.90 | 94.38 | **95.18** |
| qnli | **89.01** | 88.82 | 88.87 |
| qqp | **87.06** | 85.07 | 86.95 |
| mnli | 80.85 | **85.75** | 85.48 |
| mnli (mm) | 82.90 | **86.25** | 86.22 |
| **Avg.** | **77.93** | **79.92** | **78.53** |

Table 6: A side-by-side comparison of the performances of a vanilla LLAMA-2-7B model (Baseline), one augmented with the P-32 method (P-32) and one using a simple depth-wise average pooling (Avg.Pool), trained on various percentages of the stsb dataset.

As can be seen in Table 6 applying depth-wise average pooling does improve the average performance over the baseline, but falls short of achieving the same improvement as the P-32 method does. Thus, having a learned depth-wise pooling is an important component of the model.

## 7 CONCLUSION

In this paper, we introduced an innovative method aimed at enhancing the fine-tuning performance of Large Language Models (LLMs) for classification tasks. Our approach, characterized by the incorporation of a single learnable parameter for each transformer block, has demonstrated a significant improvement in performance across various models and datasets. Through extensive experiments, we established a consistent uptrend in performance metrics, with a notable 6 percentage points elevation observed in one of the tested LLMs.

The adaptability of our approach was tested across different model sizes and demonstrated compatibility with existing fine-tuning techniques focused on computational and storage efficiency. The proposed method proved particularly effective under conditions of data scarcity, showcasing its potential in practical, real-world scenarios where data limitations are a common challenge.

Given these substantial performance gains and the minimal architectural augmentations required, we advocate for the adoption of our proposed method by practitioners engaged in fine-tuning LLMs for classification. This approach promises not only enhanced performance but also offers insights into the depth-wise information processing, potentially paving the way for future advancements in the field of natural language processing.

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

## A ADDITIONAL ANALYSIS

| Num Blocks | 5 | 10 | 15 | 20 | 25 | 30 |
|---|---|---|---|---|---|---|
| cola | 20.99 | 46.11 | 46.73 | **49.23** | 36.69 | 47.25 |
| mnli | 71.85 | 81.4 | 85.84 | 85.17 | **85.9** | 84.42 |
| mnli (mm) | 71.75 | 82.06 | 86.42 | 86.49 | **86.54** | 85.1 |
| mrpc | 77.52 | 82.02 | 84.45 | 81.07 | 80.36 | **84.5** |
| qnli | 75.93 | 86.89 | **89.69** | 89.6 | 88.34 | 88.94 |
| qqp | 81.54 | 85.43 | 86.66 | 86.57 | 86.73 | **86.88** |
| rte | 52.71 | 56.32 | 54.51 | **62.82** | 48.74 | 52.34 |
| sst2 | 86.35 | 93.23 | 94.15 | **95.07** | 94.61 | 94.72 |
| stsb | 78.00 | 78.66 | 85.21 | **85.71** | 85.12 | 84.27 |
| **Avg.** | **68.52** | **76.90** | **79.30** | **80.19** | **77.00** | **78.71** |

Table 7: The performance achieved by an LLAMA-2-7B model when only using the first n transformer blocks. Scores are reported as Accuracy for all, except mrpc (F1), qqp (F1) and stsb (Spearman-Correlation).

Figure 4 is the visualization of the data shown in Table 4, which we also re-print here for ease of comparison (Table 7).

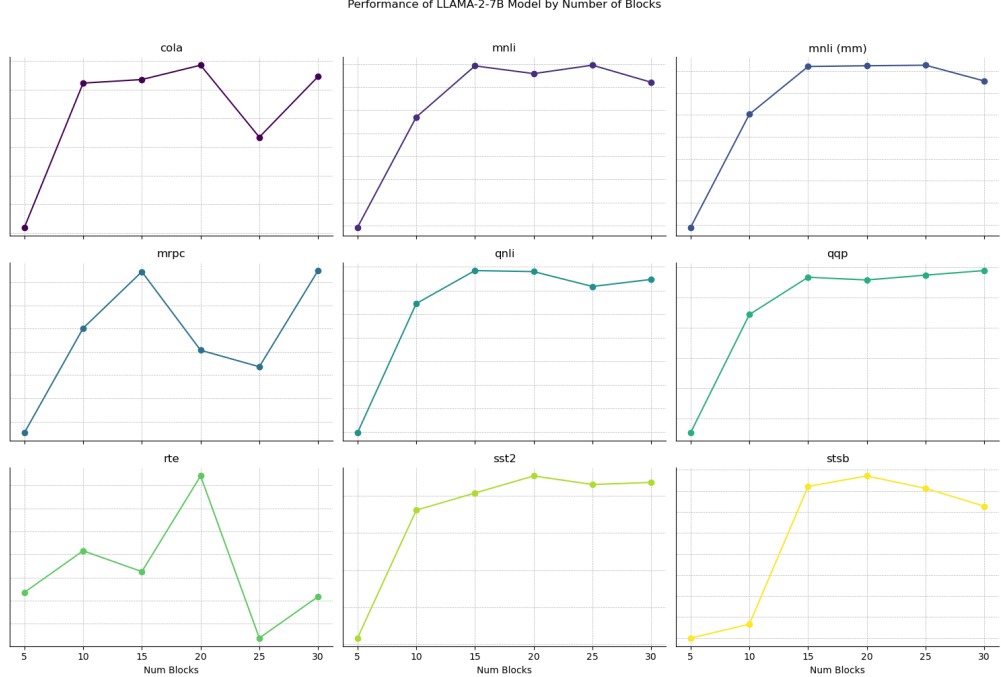

Figure 4: The results from Table 4 visualized.

