# OpenReview forum: "Adding 32 Parameters to a LLM can improve fine-tuned classification performance by up to 1.5-6 percentage points"
_ICLR.cc/2024/Conference — ICLR 2024 Conference Withdrawn Submission_

### Official Review · Reviewer_K9mr · 2023-10-27

**Soundness:** 1 poor
**Presentation:** 2 fair
**Contribution:** 1 poor
**Rating:** 3
**Confidence:** 4

**Summary:**

The paper presents a simple approach to improve fine-tuned classification performance using Large Language Models (LLMs). The proposed method, P-32, adds one learnable scalar value to each layer in an LLM serving as the weight for depth-wise pooling. The resulting method enhances the classification accuracy across the GLUE classification tasks compared to the baseline method.

**Strengths:**

* The proposed method is very simple to use in practice.
* The proposed method demonstrates some improvements over the baseline method (however, the evaluation is largely problematic, see weaknesses below).

**Weaknesses:**

* Unclear and insufficient baseline comparisons: The paper only compares the proposed method to one baseline setting, and the baseline method setup is very unclear -- does it refer to full-parameter fine-tuning of the entire LLM? Also, the baselines should include parameter-efficient fine-tuning methods like LoRA, prefix-tuning, adapters, etc.
* Inappropriate choice of evaluation tasks/settings: The current (autoregressive) LLMs, when used for classification tasks, are either via in-context learning or fine-tuned to generate the label tokens. However, this paper adds another classification head and fine-tunes it upon the hidden representations -- this is the standard practice for BERT-style models (which are not strictly "LLMs") but is no longer the common approach for fine-tuning autoregressive LLMs. In addition, the current LLMs are more suitable to be used in generation tasks than classification tasks (because BERT-style models are more parameter-efficient than LLMs for these tasks), and I expect the authors to evaluate the LLMs' performance more on generation tasks in order to substantiate the claim that their proposed approach is applicable to LLMs. Overall, the majority of the evaluation results (for non-BERT models) are not practically meaningful for LLM research.

**Questions:**

Could you clarify the baseline method setup?

---

### Official Review · Reviewer_NqDp · 2023-10-29

**Soundness:** 2 fair
**Presentation:** 3 good
**Contribution:** 2 fair
**Rating:** 3
**Confidence:** 4

**Summary:**

This paper presents an interesting way to improve the fine-tuning performance by only adding a weighting parameter for each transformer layer in PLMs. The basic idea is that different layers in PLMs encode different knowledge and different downstream tasks would require different levels of knowledge. Therefore, especially for deep PLMs, it is very likely to have the so-called “overthinking” problem. With the proposed P-32 approach, the authors show an improvement in the fine-tuning performance of GLUE for BERT, StableCode, and LLAMDA. The authors also performed comprehensive comparative studies to understand the learned weights for different layers and the sensitivity in terms of dataset size and dataset type. The comparison with average depth pooling also justifies the benefit of using learnable weights.

**Strengths:**

1. Overall the paper is written clearly and easy to follow
2. There are abundant ablation studies trying to understand and interpret the learned weights of P-32.

**Weaknesses:**

1. The experimental setup does not seem to be very reasonable. For example, StableCode is trained on programming languages but the paper is using it on NLU tasks.
2. The evaluation is only based on GLUE tasks, while the community typically would not use GLUE to evaluate models PLMs like LLAMA and StableCode. However, the improvement is only observed when finetuning these two models.
3. The novelty is a bit limited. P-32 is a simple extension to depth-wise average pooling, which is already widely known in the community.
4. While only 32 parameters are added, there is actually no computation and storage saved since we will still need to pay the full cost to compute the gradients of the whole model and save all the weights if we have multiple downstream tasks. This may limit the impact of this work.

**Questions:**

1. By comparing Table 2 and Table 4, we can see that P-32 cannot achieve similar performance compared with just using the first several layers of LLAMA on some datasets, e.g. for RTE. However, since P-32 learns weights for each layer, it should easily learn a weight vector that assigns all layers after 20 to be zero. But P-32 underperforms the first 20 layers in a few cases. This seems to cast some doubts on the effectiveness of P-32.
2. Is there any specific reason or justification for selecting these LLMs? If it is claimed that the network depth is a key factor regarding whether P-32 would be effective, I would at least expect BERT-large to be included in experiments, which contains 24 layers and is more reasonable to compare with BERT-base. Comparing BERT-base and LLAMA is not very convincing because there are a lot more differences between the two PLMs.

---

### Official Review · Reviewer_XMXP · 2023-11-01

**Soundness:** 3 good
**Presentation:** 3 good
**Contribution:** 3 good
**Rating:** 5
**Confidence:** 3

**Summary:**

This paper proposes a method to fuse the representations of each transformer block via weighted pooling to obtain the sentence representation, which alleviates the overthinking problem of deep language models. For the Llama2-7b model, adding only 32 parameters can significantly improve its performance on various natural language classification tasks. Moreover, the experimental results demonstrate the efficacy of the proposed method, and an ablation study confirms the superiority of the weighted pooling over the simple average pooling.

**Strengths:**

1. Adapting public large language models such as Llama2 to domain-specific tasks is important for enhancing their performance. Fine-tuning is often essential to achieve good results. This paper proposes a simple method to mitigate the overthinking problem during fine-tuning. The empirical results show that the proposed weighted pooling method can alleviate this phenomenon, thus improving the performance.

2. Extensive analysis and ablation experiments confirm the existence of the overthinking issue and validate the efficacy of the proposed method. The correlation analysis of multiple dimensions also offers useful insights into the proposed method.

3. The method is straightforward and technically sound.

**Weaknesses:**

There are some concerns are not addressed in the paper.

1. From the results in Table 2, the Llama-2-13B model does not benefit more from the weighted pooling than the Llama-2-7B model. The paper does not explain this reason well. Intuitively, the 13B model is deeper than the 7B model, and it should have a larger performance improvement.

2. It seems the proposed method can also enhance the performance of LLMs on generation tasks, which may also suffer from the overthinking issue. However, the paper lacks such analysis.

3. Although the average pooling strategy can improve the performance, the results in Table 4 show that it cannot learn the optimal layers. For example, the best performance for RTE is 62.82 when tuning the first 20 layers, but it drops to 57.04 when using weighted pooling. This indicates that there is still room for improvement for the weighted pooling method.

**Questions:**

1. How does the proposed method affect the performance of generation tasks?

2. When fine-tuning the LLMs with a parameter-efficient method like LoRA, does the proposed method still provide benefits?

---

### Official Review · Reviewer_Th33 · 2023-11-10

**Soundness:** 3 good
**Presentation:** 3 good
**Contribution:** 1 poor
**Rating:** 3
**Confidence:** 3

**Summary:**

This paper proposes P-32, an architectural augmentation to improve classification performance by fully utilizing the hierarchical information captured by the pre-trained Large Language Model (LLM) that is - (i) able to select and retain task-relevant information
captured at different depths in the LLM, (ii) improves performance across a plethora of different model sizes and in tandem with popular fine-tuning techniques, and (iii) specifically and successfully addresses the problem of overthinking. Towards this, authors have leveraged weighted pooling over the average pooling and fused the pooled representation, encapsulating features from various depths
to the model through the standard classification head. The proposed approach shows an improvement over GLUE for BERT, StableCode, and LLAMA. Authors have also done an ablation study that validates the effectiveness of weighted pooling.

**Strengths:**

1. The paper is easy to follow.
2. Improvement in performance with up to 6 percentage points elevation observed in one of the tested LLMs.
3. The proposed method is effective under conditions of data scarcity again, which seems a real-life practical scenario in many cases.
4. The correlation analysis study offers valuable insights.

**Weaknesses:**

1. Table 2 shows that LLAMA-2-13B (deeper model) performs poorer than LLAMA-2-7B. A proper reasoning is missing behind this
2. The LLM trained over programming languages (StableCode) is being used for NLU tasks.
3. The study must have considered some PaLM: Scaling Language Modeling with Pathways (Chowdhery et al.), which is trained on corpus with a wide range of natural language use cases.
4. The study should validate the performance of P-32 on generation tasks.

**Questions:**

1. Can we consider BERT as LLM today?